# In-Situ Seawater Gamma Spectrometry with LaBr₃ Detector at a Nuclear Power Plant Outlet

Daowei Dou [1], Zhi Zeng [1], Wen Yu [2,*], Ming Zeng [1], Wu Men [3], Feng Lin [3], Hao Ma [1], Jianping Cheng [4] and Junli Li [1]

1 Department of Engineering Physics, Tsinghua University, Beijing 100084, China; ddw15@mails.tsinghua.edu.cn (D.D.); zengzhi@tsinghua.edu.cn (Z.Z.); zengming@tsinghua.edu.cn (M.Z.); mahao@tsinghua.edu.cn (H.M.); lijunli@tsinghua.edu.cn (J.L.)
2 School of National Security and Emergency Management, Beijing Normal University, Zhuhai 519087, China
3 Third Institute of Oceanography, Ministry of Natural Resources, Xiamen 361005, China; menwu@tio.org.cn (W.M.); linfeng@tio.org.cn (F.L.)
4 College of Nuclear Science and Technology, Beijing Normal University, Beijing 100875, China; chengjp@tsinghua.edu.cn
* Correspondence: yuw@bnu.edu.cn

**Abstract:** In a nuclear emergency, the application of in situ spectrometers for monitoring environmental radioactivity is significantly important, as information on the type and activity of radionuclides released from the accident can be obtained quickly. However, in emergency environmental radiological monitoring, a balance between energy resolution and detecting efficiency must be considered in selecting an appropriate detector. In this study, in situ gamma spectrometry was conducted with the LaBr₃ detector to determine the radioactivity of seawater at the discharging outlet of a coastal nuclear power plant in southeast China. The results show that the LaBr₃ scintillator has excellent energy resolution and detection efficiency, making it a promising detector for emergency monitoring.

**Keywords:** LaBr₃ (Lanthanum bromide); In situ gamma-ray spectrometry; environmental radioactivity; Monte Carlo simulation; nuclear power plant effluents

## 1. Introduction

The Fukushima nuclear accident has increased the importance of in-situ spectrometry. It can obtain necessary information such as the type and activity of nuclides released in an accident within a short period, which can help in decision making and source term evaluation in case of an accident. In addition, the regulatory authorities currently take samples from the waste tank to determine whether the discharge of liquid radioactive substances complies with the regulations. The environmental regulatory authorities mainly obtain seawater, sediment, and organism samples from the surrounding sea area for analysis 2-4 times a year to judge whether the environmental radioactivity is in compliance with the standards. From the public's perspective, there is a desire for more timely access to more detailed information. From the regulatory perspective, there is a desire for easier monitoring of radioactive effluents.

The development of in situ γ-energy spectrometry in seawater was first pioneered around 1950, and some countries have achieved automatic online measurement of ocean γ-energy spectra. The Marine Environment Laboratory of International Atomic Energy Agency (IAEA-MEL) developed a dual-probe marine in situ γ spectrometer using a NaI(Tl) scintillator with high detection efficiency and a High Purity Germanium (HPGe) semiconductor with high energy resolution to investigate artificial radionuclide contamination in a variety of marine environments. The device was applied to an environmental radioactivity survey at a nuclear waste disposal site in the Kara Sea and a seafloor sediments γ-ray survey in the Irish Sea near the Sellafield nuclear fuel reprocessing plant in the UK to obtain the distribution and inventory of ¹³⁷Cs [1–4]. POSEIDON, a marine γ-radiation monitoring

network consisting of 11 ocean observation buoys and a control center, was established in Greece. The buoys transmit data collected by NaI(Tl) detectors to the Hellenic Marine Research Center through satellite communication or GSM cell phone communication, and the single acquisition time of the spectrometer system is 3 h [5–8]. The German Federal Ministry of the Environment established a monitoring network including 13 fixed stations at sea and on the seashore to monitor radioactive contamination in the German Gulf and the Western Baltic Sea. The sea stations are set up on buoys, and the seaside stations on tide gauges. In addition, four sets of equipment are installed on vessels of the Federal Maritime and Hydrographic Agency of Germany (BSH) to inspect hotspots of artificial radioactivity when needed. Gamma-ray detection in seawater is carried out from 2 to 6 m below the water surface. The data recorded by fixed stations are transmitted to the central computer of BSH via satellite (offshore stations) or telephone (seaside stations) [9]. The Kurchatov Institute of the Russian Research Center developed the REM-10 series of highly sensitive underwater $\gamma$-energy spectrometers based on NaI(Tl) scintillators for effective monitoring of radioactive contamination of various waters. This series of underwater gamma energy spectrometers were used in the investigation of the wrecked nuclear submarines "Komsomolets" and "Kursk" as well as at two radioactive waste dumping sites in the Kara Sea and Novaya Zemlya bays [10]. In China, a seawater radioactivity monitoring device was developed using a NaI(Tl) scintillator, which can be suspended either on a buoy for fixed-point measurements or on the bottom of a vessel for cruising monitoring, with an energy resolution of 14.8% for $^{137}$Cs (at 662 keV) [11].

From the above introduction, it can be seen that NaI(Tl) detectors are commonly used in seawater in-situ $\gamma$ spectrometers. NaI(Tl) detectors have the advantages of high detection efficiency, wide applicable temperature range, and low price. Nevertheless, it also has an apparent deficiency of low energy resolution. Typically NaI(Tl) detectors can only achieve an energy resolution as good as 7.5%, and the energy spectrum peak position drifts during long-time operation. Therefore, the NaI(Tl)-detector-based seawater in situ $\gamma$ spectrometer has difficulties in qualitative judgment and quantitative measurement when encountering complex $\gamma$ spectrum. As concerns progress for applying underwater medium resolution gamma-ray spectrometer in the marine environment, a recent detection system is developed (named GeoMAREA) integrating a 2″ × 2″ CeBr$_3$ crystal [12]. This system is applied effectively for studying groundwater-seawater interaction in submarine groundwater discharge system [13,14]. Furthermore, the advantage of the HPGe detector is the good energy resolution, which makes it has excellent ability of peak discrimination and radionuclide identification. However, the HPGe system is not very robust. It needs to operate at a low temperature, so it consumes more power compared to low and medium resolution systems and it cannot detect continuously for a long time. Therefore, if there is any detector with high detection efficiency and good energy resolution at room temperature, it would be an optimum system for in situ $\gamma$-ray spectrometry and long-term monitoring in seawater. In this regard, an attempt has been made in this study to obtain in-situ gamma spectrometry of liquid discharges by deploying a lanthanum bromide scintillator fixed to the buoy at the outlet of a coastal nuclear power plant.

## 2. Materials and Methods

### 2.1. LaBr$_3$ Scintillator

One 2″φ × 2″ LaBr$_3$ scintillator (Huakailong®, Beijing, China) and a photomultiplier tube (PMT) connected to it were encapsulated together in a 0.5 mm thick aluminum alloy case. Then, the other electronic modules were encapsulated together with this aluminum alloy case in a waterproof PVC container. At the experiment site, the PVC container is suspended below the sea surface by a buoy. It collects data in-situ and transmits them to the land base via Global System for Mobile Communication (GSM). Figure 1 shows the external appearances of the aluminum alloy case and the PVC container of the detector.

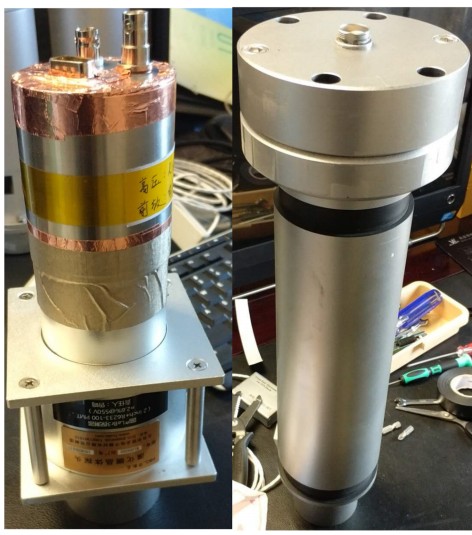

**Figure 1.** External appearances of the aluminum alloy case and the PVC container of the detector.

*2.2. Experiment Site*

Fujian Province is located on the southeast coast of China and has a shortage of conventional energy resources. In recent years, with the rapid rise in demand for energy as the economy develops, it is imperative to develop nuclear energy vigorously. As of May 2021, four nuclear power plants have been built in Fujian Province, including Ningde, Fuqing, Xiapu, and Zhangzhou. Ningde Nuclear Power Plant is located in Taimushan Town, Fuding City (Figure 2) and is planned to construct six megawatt-class pressurized water reactor nuclear power units, with four units to be built in the first phase of the project using CPR1000 technology.

The buoy with the detector is positioned 100 m southeast of the discharging outlet of Ningde Nuclear Power Station, and the water depth there is approximately 5 m to 15 m. The detector is suspended at a depth of 1.5 m from the water surface, and there is no risk of bottoming out at low tide.

The buoy consists of two parts. The upper part is equipped with solar panels and antennas for powering the equipment and transmitting signals, respectively. The lower part is a cabin structure that houses the supporting equipment. The detector is fixed to the bottom of the buoy by a bracket. The bottom of the buoy is connected to an anchor fixed to the seabed.

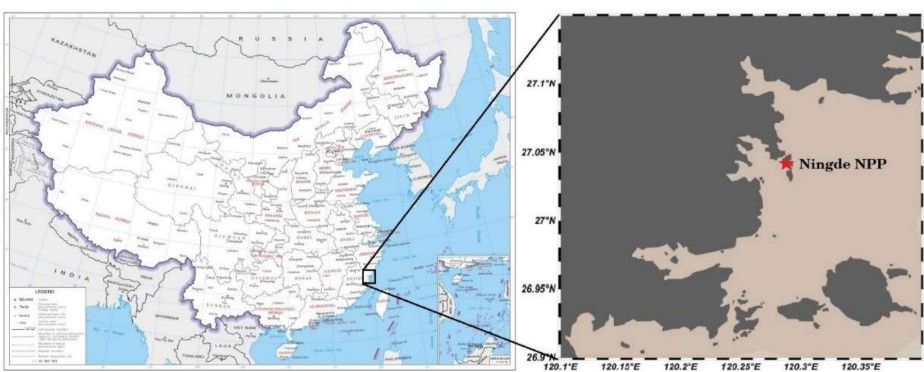

**Figure 2.** Location of the experiment site.

*2.3. Radionuclides in Liquid Effluents from Nuclear Power Plants*

The primary radionuclides discharged from Ningde NPP in liquid form are $^{110m}$Ag, $^{51}$Cr, $^{54}$Mn, $^{137}$Cs, $^{134}$Cs, and $^{124}$Sb. The full energy peak count rate n of a particular $\gamma$-ray is proportional to the product of the emission probability $I_E$ and the source peak detection

efficiency $\varepsilon_E$. Therefore, for each monitoring radionuclide, the $\gamma$-ray with the largest $I_E \times \varepsilon_E$ should be selected.

For radionuclides that emit only one type of $\gamma$-ray, such as ${}^{51}$Cr, ${}^{54}$Mn, ${}^{137}$Cs, that ray is chosen as the characteristic $\gamma$-ray. Since that the $\gamma$-ray with energy 602.726 keV for ${}^{124}$Sb and the $\gamma$-ray with energy 604.721 keV for ${}^{134}$Cs are very close and may overlap in the spectrum, the $\gamma$-rays with energies 1690.971 keV and 795.864 keV were selected in the energy spectrum processing as the characteristic $\gamma$-rays for ${}^{124}$Sb and ${}^{134}$Cs respectively to calculate their activity concentrations. ${}^{137}$Cs $\gamma$-rays with the energy of 661.657 keV and ${}^{110m}$Ag $\gamma$-rays with the energy of 657.76 keV are also closer together in the spectrum, but ${}^{137}$Cs emits only one major $\gamma$-ray, so the energy of 884.678 keV was chosen as the characteristic $\gamma$-ray of ${}^{110m}$Ag in the process of spectrum processing.

## 3. Results

### 3.1. Calibration

#### 3.1.1. Energy Calibration

The purpose of the energy calibration is to determine the relationship between the particle energy and the channel position in the PMT for subsequent analysis of the obtained energy spectrum. The LaBr$_3$ crystals naturally contain the radioactive isotope ${}^{138}$La. In addition, because the lanthanides are chemically similar to the actinides and cannot be easily separated, a certain amount of the radionuclide ${}^{227}$Ac is also present within LaBr$_3$ crystals. Both ${}^{138}$La and ${}^{227}$Ac contribute to the intrinsic radiation of the LaBr$_3$ crystal and form peaks in the background energy spectrometry. However, these peaks do not overlap with the peaks of the primary radionuclides discharged from Ningde NPP in the liquid form listed in Section 2.3. In the meantime, the peaks of the LaBr$_3$ intrinsic radiation can help perform energy calibration of the detector and self-stabilizing spectrum correction.

The effect of ${}^{227}$Ac on the background spectrum is mainly in the high-energy end, and it arises from a series of alpha decays of its parent and daughters. There are two pathways of decay for ${}^{138}$La, one by electron capture to generate ${}^{138}$Ba and the other by beta decay to ${}^{138}$Ce. The process of electron capture is accompanied by the emission of characteristic X-rays, with a 63.7% probability of emitting K$_\alpha$ X-rays at 31.84 keV, followed by the Osher effect to generate a 5.6-keV oscillator electron; a 27.5% probability of L$_\alpha$ X-rays with an energy of 4.5 keV; and an 8.8% probability of M$_\alpha$ X-rays with an energy of about 1 keV. The Osher electrons and X-rays will undergo a coincidence summing, forming a characteristic X-ray peak in the spectrum with an energy of about 36 keV. ${}^{138}$Ba emits $\gamma$-rays with an energy of 1436 keV when it transits from the excited to the steady state and undergoes a coincidence summing with the characteristic X-rays, forming characteristic peaks in the spectrum with energies of 1441 keV and 1472 keV, respectively. During the decay of ${}^{138}$La to form ${}^{138}$Ce, $\beta$-electrons are generated with a maximum energy of 255 keV. The excited state of ${}^{138}$Ce leaps to the steady state and emits $\gamma$-rays with an energy of 789 keV. The $\beta$ electrons and $\gamma$ rays also undergo coincidence summing and form a continuous spectrum in the spectrum with energies of 789~1044 keV. Based on the above analysis, the 36 keV, 1441 keV, and 1472 keV characteristic peaks of ${}^{138}$La were selected to calibrate the energy-channel position relationship.

#### 3.1.2. Efficiency Calibration

Spiked Experiment

If the activity concentration of a radionuclide in seawater of volume $V$ (in L) is $A$ (in Bq L$^{-1}$), the probability of emission of a particular $\gamma$-ray from that radionuclide is $I_E$, and the detector's efficiency in detecting the source peak of that $\gamma$-ray is $\varepsilon_{sp}$, then the detector's full-energy peak count rate $n$ (in cps, counts per second) for that $\gamma$-ray should theoretically be $n = A V I_E \varepsilon_{sp}$. Specific count rate $y = V I_E \varepsilon_{sp}$ (in cps L Bq$^{-1}$) represents the full energy peak count rate of a particular $\gamma$-ray emitted by the radionuclide of unit activity concentration in seawater, which is the detector's detection efficiency for that $\gamma$-ray.

The efficiency calibration experiment was performed in two water tanks with a diameter of 2.0 m and a height of 2.3 m to simulate an underwater environment. One of the tanks was spiked with a standard solution of $^{137}$Cs to make the activity concentration of $^{137}$Cs in the tank to be 0.45 Bq/L, and the other tank was spiked with $U_3O_8$, making the activity concentration of $^{238}$U in the water tank to be 5.86 Bq/L. Since the half-life of $^{238}$U ($4.47 \times 10^9$ a) is much longer than the half-lives of its decay daughters $^{234}$Th ($T_{1/2} = 24.1$ d) and $^{234m}$Pa ($T_{1/2} = 1.17$m), this decay system established long-term equilibrium within a few months, and the $^{234}$Th activity and $^{238}$U activity were almost identical so that the activity of $^{238}$U can be reflected by the γ-rays emitted by $^{234}$Th at 92.38 keV (2.18%) and 92.80 keV (2.15%).

In the experiment, the measurement times for $^{137}$Cs and $^{234}$Th were 118.8 h and 41.1 h, respectively, and the results are shown in Table 1.

**Table 1.** Measurement results in efficiency calibration with spiked seawater.

| Radionuclides | $^{137}$Cs | $^{234}$Th |
|---|---|---|
| Activity concentration in tank water (Bq/L) | 0.45 | 5.86 |
| full energy peak count rate (cps) | 0.036 | 0.037 |
| detection efficiency (cps L Bq-1) | 0.080 | 0.006 |

Monte-Carlo Simulation

The detection efficiencies of $LaBr_3$ for thirteen radionuclides were also obtained using the Monte Carlo simulation method. Considering that seawater is a good shield for radiation, the contribution of radioactive sources located more than 1 m from the detector is not considered in the study. As the distance between the radioactive source and the detector crystal varies, its contribution to the total energy peak count also varies. Therefore, the simulation was carried out using a sub-shell sampling method for every 5 cm thick spherical shell with a total shell thickness of 100 cm, and finally, the detection efficiency in the total simulated volume can be obtained by weighted summing according to the volume of each layer to the total volume. The simulation results for each nuclide are shown in Table 2. The detection efficiency for $^{137}$Cs at the 662 keV peak of the device developed in this study is 0.0867 cps/Bq $L^{-1}$, very close to the efficiency of $CeBr_3$ detector, another medium-resolution system [12].

**Table 2.** Detection efficiency of different radionuclides obtained with Monte Carlo simulation.

| Radionuclide | γ-ray Energy (keV) | Emission Probability (%) | Detection Efficiency (cps/Bq $L^{-1}$) |
|---|---|---|---|
| $^{234}$Th | 92-93 | 4.33 | 0.0062 |
| $^{131}$I | 364.49 | 81.20 | 0.1113 |
| $^{134}$Cs | 569.331 | 15.37 | 0.0165 |
| $^{124}$Sb | 602.728 | 97.78 | 0.1013 |
| $^{134}$Cs | 604.722 | 97.63 | 0.1005 |
| $^{110m}$Ag | 657.75 | 94.38 | 0.0952 |
| $^{137}$Cs | 661.659 | 84.99 | 0.0867 |
| $^{134}$Cs | 795.868 | 85.47 | 0.0791 |
| $^{58}$Co | 810.766 | 99.44 | 0.0924 |
| $^{54}$Mn | 834.855 | 99.97 | 0.0932 |
| $^{110m}$Ag | 884.67 | 74.00 | 0.0680 |
| $^{60}$Co | 1173.24 | 99.85 | 0.0845 |
| $^{60}$Co | 1332.508 | 99.98 | 0.0768 |
| $^{40}$K | 1460.822 | 10.55 | 0.0079 |
| $^{124}$Sb | 1690.984 | 47.46 | 0.0348 |
| $^{208}$Tl | 2614.511 | 99.76 | 0.0590 |

To eliminate the effect of emission probability, we normalized the probability of γ-ray emission for all energies. The relationship between detection efficiency and energy was

plotted as shown in Figure 3. The normalized measured values of detection efficiency and the Monte Carlo simulated values are plotted together, as shown in Figure 3.

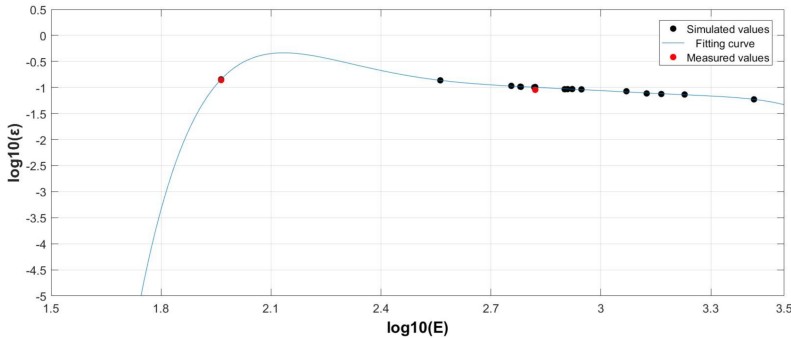

**Figure 3.** Comparison of normalized detection efficiency measured values and Monte Carlo simulated values.

### 3.1.3. Energy Resolution

The energy resolution is an important parameter characterizing the detector's ability to distinguish the energy of the detected particles. The detector's energy resolution in this study was determined by counting standard point sources of $^{60}$Co, $^{137}$Cs, $^{152}$Eu, and $^{241}$Am for 1000 s, and the full width at half maximum (FWHM) of each γ-ray are shown in Table 3. The fitted curve for energy resolution η (%) versus energy E (keV) is:

$$\eta = 1.088 \times E^{-0.5788} \tag{1}$$

The typical energy resolution for $^{137}$Cs of NaI detectors, which are widely used in in-situ monitoring in marine nuclear emergency scenarios, is about 6–9% [6,10,15,16]. The energy resolution for $^{137}$Cs of the novel cerium bromide (CeBr$_3$) detector is ~3.5% [12]. The device developed in this study has an energy resolution for $^{137}$Cs of 2.60%, which is significantly better than that of the NaI detector and comparable with the CeBr$_3$ detector. Therefore, the application of medium resolution systems such as LaBr$_3$ and Ce Br$_3$ detectors will benefit the rapid identification of various radionuclides in nuclear emergency scenarios.

**Table 3.** Full width at half maximum (FWHM) of four radionuclides.

| Radionuclide | γ-ray Energy (keV) | Emission Probability (%) | FWHM (%) |
|:---:|:---:|:---:|:---:|
| $^{60}$Co | 1173.23 | 99.85 | 1.93 |
| | 1332.49 | 99.98 | 1.82 |
| $^{137}$Cs | 661.66 | 85.1 | 2.60 |
| $^{241}$Am | 59.54 | 35.9 | 10.22 |
| $^{152}$Eu | 121.78 | 28.53 | 6.71 |
| | 344.28 | 26.59 | 3.53 |
| | 964.06 | 14.51 | 2.01 |
| | 1112.08 | 13.67 | 1.99 |
| | 1408.01 | 20.87 | 1.49 |

### 3.2. Self-Stabilizing Spectrum Correction

Due to factors such as changes in ambient temperature, the measured energy spectrum is affected by spectral drift, i.e., changes in the channel positions of γ-rays of the same energy on the spectrum. In order to solve the problem of energy spectrum drift, a quadratic polynomial relationship between the energy and the channel position of the standard and actually measured spectrum is obtained using the characteristic peaks of the detector crystal naturally occurring radioactivity at 36 keV, 1472 keV, and 2635 keV, which enables the self-stabilizing spectrum correction, shown in Figure 4.

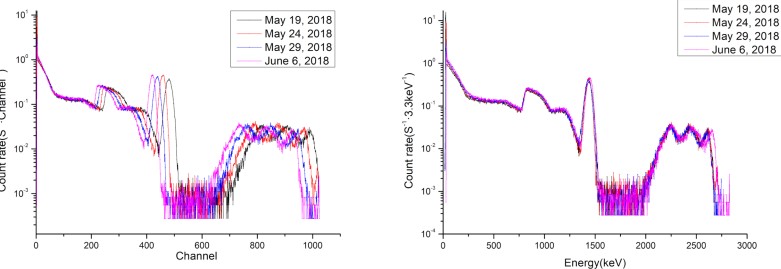

**Figure 4.** Spectrums before (left) and after (right) self-stabilizing correction.

### 3.3. Minimum Detectable Activity Concentration (MDAC)

The minimum detectable activity concentration (MDAC) is an essential parameter of the detector system, which represents the lowest activity concentration of a radionuclide that can be determined to be present in a sample at a certain confidence level at a specific measurement time. At a confidence level of 95%, MDAC can be expressed as:

$$\text{MDAC} = \frac{L_D}{tI_E\varepsilon_{sp}V} = \frac{2.71 + 4.65\sqrt{N_b}}{tI_E\varepsilon_{sp}V} \tag{2}$$

where $N_b$ refers to the background count, $t$ refers to the live detection time, $I_E$ refers to the emissivity of the corresponding $\gamma$-ray, $\varepsilon_{sp}$ refers to the detection efficiency of the detector for that $\gamma$-ray, and $V$ refers to the detection volume of the detector. For the background count $N_b$, $C - 3\sigma$ is used as the left marker of the region of interest (ROI), and $C + 3\sigma$ is used as the righte marker of the ROI. Here, $C$ is the channel location for the energy of the characteristic peak, $\sigma$ is the standard deviation of the peak that fits a normal distribution. At the measurement time of 120 h and 1 h, the MDACs of the device developed in the study for targeting radionuclides are listed in Table 4. The MDAC for $^{137}$Cs in 1 h is 1.10 Bq/L, which is higher than that of NaI detector (approximately 0.2 Bq/L [6,17,18]). However, considering that the radioactivity in the seawater adjacent to the NPP will be fairly high in the event of an accident [19–21], the MDACs of the device developed in this study will be low enough to quantify the activity of radionuclides released from the accident.

**Table 4.** Minimum detectable activity concentrations of targeting radionuclides.

| Radionuclide | $\gamma$-ray Energy (keV) | Emission Probability (%) | MDAC in 120 h (Bq L$^{-1}$) | MDAC in 1 h (Bq L$^{-1}$) |
|---|---|---|---|---|
| $^{54}$Mn | 834.848 | 0.09998 | 0.150 | 1.63 |
| $^{51}$Cr | 320.0824 | 0.0991 | 0.586 | 6.39 |
| $^{137}$Cs | 661.657 | 0.851 | 0.101 | 1.10 |
| $^{124}$Sb | 602.726 | 0.9779 | 0.144 | 1.57 |
| | 1690.971 | 0.4757 | 0.018 | 0.20 |
| $^{134}$Cs | 604.721 | 0.9762 | 0.092 | 1.00 |
| | 795.864 | 0.8546 | 0.157 | 1.71 |
| $^{110m}$Ag | 657.76 | 0.9561 | 0.092 | 1.00 |
| | 884.678 | 0.7496 | 0.196 | 2.14 |

### 3.4. In-Situ Continuous Observation Results

From April to August 2018, the accumulative seawater gamma spectrum was collected using the LaBr$_3$ underwater detector mounted on a buoy at the discharging outlet of Ningde NPP. During the monitoring process, the spectrum data was automatically saved every 5 min to ensure the continuity and integrity of the data. Afterward, self-stabilization correction was performed, and then the energy spectrum was superimposed to obtain the accumulated spectrum under long-term measurement, which was used as the spectrum for data analysis (Figure 5).

The characteristic peaks found in the energy spectrum are mainly 351 keV and 609 keV, besides the characteristic peak of lanthanum bromide itself. There are a few possible sources for the 609 keV peak: the 609 keV characteristic γ-ray of the natural radionuclide $^{214}$Bi, which is one of the daughters of $^{222}$Rn and is affected by precipitation or the sea flows [22–24], or artificial radionuclides $^{124}$Sb (602.7 keV) or $^{134}$Cs (604.7 keV), which are two common radionuclides in liquid effluents from NPPs.

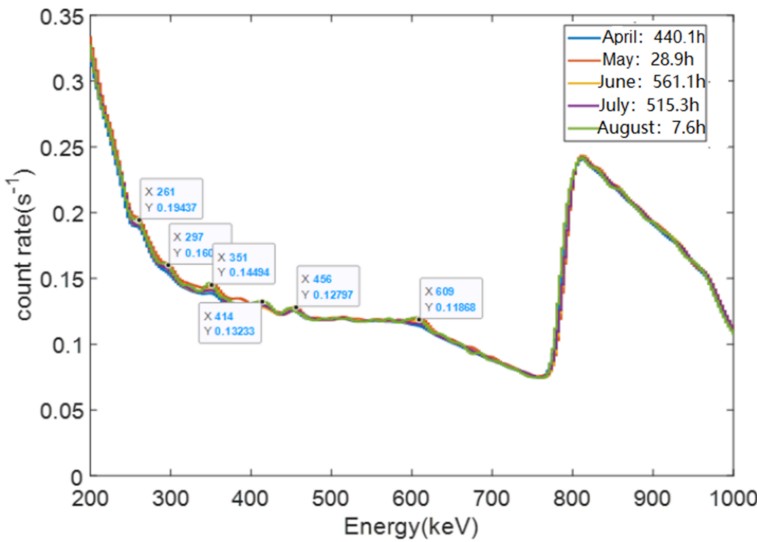

**Figure 5.** Spectrum at 200 to 1000 keV in April—August 2018.

## 4. Discussion

In order to investigate the reasons for the 609 keV peak, its daily counts were recorded. There were several days in which the peak counts showed a significant increase, and the corresponding dates are marked in Figure 6.

When it rains, radionuclides in the atmosphere and on land may be carried into the ocean by the rain, which may result in an increase in the energy spectrum counts. The precipitation during the test period is summarized in Table 5. It can be seen that most of the rainfall is accompanied by an increase in 609 keV peak counts. However, there are also some rainy days when no increase in peak counts is observed, and there are some days with no precipitation but an increase in 609 keV peak count. It can be concluded that the changes in 609 keV peak counts are related but not fully synchronized with the changes in precipitation conditions.

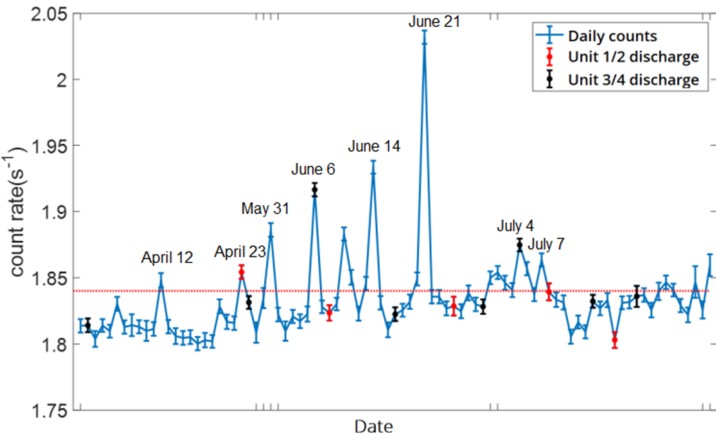

**Figure 6.** Daily counts of the peak at 609keV (red and black marks are the days with discharge from reactor units 1/2 and units 3/4, respectively).

**Table 5.** Dates with precipitation during the test period and the relevant increase in 609 keV peak count.

| Date | Weather (Day/Night) | Increase in 609 keV Peak Count? |
|---|---|---|
| April 12 | medium rain/medium rain | Yes |
| April 14 | Medium Rain/light rain | No |
| April 23 | thunderstorm/thunderstorm | Yes |
| May 31 | medium rain/heavy rain | Yes |
| June 6 | rainstorm/medium rain | Yes |
| June 14 | light rain/mostly cloudy | Yes |
| June 18 | medium rain/light rain | No |
| June 21 | heavy rain/light rain | Yes |
| July 2 | heavy rain/mostly cloudy | No |
| July 4 | mostly cloudy/mostly cloudy | Yes |
| July 7 | light rain/mostly cloudy | Yes |
| July 10 | mostly cloudy/heavy rain | No |
| July 11 | rainstorm/light rain | No |

To better clarify the source of the measured 609 keV peak, the authors recorded the liquid effluent discharges from the Ningde NPP. During the test period (April to July 2018), a total of seven discharges were conducted at units 1 and 2 and nine discharges at units 3 and 4 of the Ningde NPP. For each discharge, one tank of waste liquid with a volume of 300 $m^3$ was discharged in about 20 h. The activity levels of $^{124}$Sb were all below the detection limit (about 0.6 Bq/L) in all the discharges. Considering that the detector was deployed at a distance of 100 m from the discharging outlet of the NPP, it can be inferred that due to the dilution of seawater, the radioactivity of $^{124}$Sb at the position of the detector was lower than the MDAC of $^{124}$Sb listed in Table 4 (0.144 Bq/L). Therefore, it is unlikely that the 609 keV peak found in the spectrum was caused by the discharge of waste liquid from the NPP.

Another possible source of the 609 keV peak is $^{214}$Bi in seawater. According to the data in Table 5, the change in the count rate of the 609 keV peak may be related to, but not entirely due to, the change in precipitation, as the changes in the two are not completely synchronized. Further research is needed to ascertain the cause of the peak count fluctuations.

Although this study has not identified the specific cause of the 609 keV peak, it is certain that since this peak does not overlap with the energy peaks of the main emissions from a nuclear power plant, the existence of this peak does not affect the regular or emergency monitoring at the NPP.

## 5. Conclusions

In the context of the rapid development of nuclear power and the increasing importance of marine environmental protection, it is necessary to establish a real-time online monitoring network of radioactivity in Chinese coastal waters.

In this study, the energy scaling of the lanthanum bromide detector was carried out using its naturally occurring radioactivity. The detection efficiency of the detector for radionuclides was obtained by spiking experiments and Monte Carlo simulations, and the relationship with γ-ray energy was derived by fitting. The energy resolution of the detector was determined using standard radioactive point sources. In this study, an in situ measurement of the gamma energy spectrum of seawater at the nuclear power plant discharge outlet was also conducted in the field for four months. The results show that the lanthanum bromide detector has excellent energy resolution and detection efficiency performance and can be used at environmental temperature. It could become another choice besides sodium iodide detectors for in situ gamma energy spectrometry of seawater in regular situations and emergencies.

**Author Contributions:** Conceptualization, Z.Z. and W.Y.; methodology, D.D., M.Z. and Z.Z.; software, D.D. and M.Z.; validation, W.M., F.L., and W.Y.; formal analysis, W.Y.; investigation, W.M., F.L.; resources, J.C.; data curation, H.M.; writing—original draft preparation, D.D. and W.Y.; writing—review and editing, Z.Z. and W.Y.; visualization, D.D.; supervision, J.C., and J.L.; project administration, Z.Z.; funding acquisition, Z.Z. All authors have read and agreed to the published version of the manuscript.

**Funding:** This research was funded by the NATIONAL KEY SCIENTIFIC INSTRUMENT AND EQUIPMENT DEVELOPMENT PROJECT, grant number 2016YFF0103902 and the PUBLIC SCIENCE AND TECHNOLOGY RESEARCH FUNDS PROJECTS OF OCEAN, grant number No.201505005.

**Institutional Review Board Statement:** Not Applicable.

**Informed Consent Statement:** Not Applicable.

**Data Availability Statement:** Not Applicable.

**Conflicts of Interest:** The authors declare no conflict of interest. The funders had no role in the design of the study; in the collection, analyses, or interpretation of data; in the writing of the manuscript, or in the decision to publish the results.

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
