# Peer review of "In-Situ Seawater Gamma Spectrometry with LaBr3 Detector at a Nuclear Power Plant Outlet"

_jmse, doi:10.3390/jmse9070721_

Round 1

Reviewer 1 Report

General comments

The authors developed in situ gamma spectrometer with LaBr3 detector and measured radioactivity of seawater at the discharging outlet of a coastal nuclear power plant in southeast China. It took 118.8 hours to measure the Cs-137 concentration of 0.45 Bq/L. Please clarify what the operational detection limit would be in the event of an accident. Since monitoring systems for the accidents are important, I think that this paper is suitable for the publication after minor revision.

Specific comments

L210 Reference is needed for 7-8%. Explicitly describe the energy resolution improved by this device.

L235 Since the 4. Discussion is a discussion of the results of the newly presented continuous observations, please structure it in a way that is easy to understand.

In discussion, they conclude that the 609 keV peak is not caused by the water discharge. In other words, to the impact of BI-214? In addition to discussing the relationship to rainfall, you should also discuss the possible impact of BI. Finally, you need to state that this peak does not affect monitoring.

Reviewer 2 Report

It seems that this paper can be used as a valuable reference for understanding the in-situ gamma-ray energy spectrum in seawater. In particular, it is interesting to introduce real monitoring data for NPP and interpret the monitoring results with rainfall. I would like to just present two issues as follow.

  1. Please provide a way to determine “Nb” in equation 3. For example, how to assign the upper and lower limits of ROI for “Nb”.
  2. Please provide the measurement time for the MDAC shown in Table 4.
